# Associations between ethnicity and persistent physical and mental health symptoms experienced as part of ongoing symptomatic COVID-19

**Sindhu Bhaarrati Naidu**[1,2], **Anita Saigal**[1,2], **Amar Jitu Shah**[1,2], **Chibueze Ogbonnaya**[3], **Shiuli Bhattacharyya**[1,2], **Karthig Thillaivasan**[1,2], **Songyuan Xiao**[1,2], **Camila Nagoda Niklewicz**[1,2], **George Seligmann**[1,2], **Heba Majed Bintalib**[2,4,5], **John Robert Hurst**[1,2], **Marc Caeroos Isaac Lipman**[1,2], **Swapna Mandal**[1,2]*

1 Respiratory Medicine, Royal Free London NHS Foundation Trust, London, United Kingdom, 2 UCL Respiratory, University College London, London, United Kingdom, 3 Institute of Child Health, University College London, London, United Kingdom, 4 Department of Respiratory Care, King Saud bin Abdulaziz University for Health Sciences, Jeddah, Saudi Arabia, 5 King Abdullah International Medical Research Centre, Jeddah, Saudi Arabia

* swapnamandal@nhs.net

**Data Availability Statement:** There are ethical restrictions in sharing a de-identified data set as our data consists of a small number (<1000) of

## Abstract

### Introduction

Ethnicity can influence susceptibility to SARS-CoV-2 infection, hospitalisation and death. Its association with ongoing symptomatic COVID-19 is unclear. We assessed if, among a population followed up after discharge from hospital with COVID-19, adults from Asian, black, mixed and other backgrounds are at increased risk of physical and mental health symptoms.

### Methods

Adults discharged after hospitalisation with COVID-19 between 03/03/2020 and 27/11/2021 were routinely offered follow-up six to 12 weeks post-discharge and reviewed for ongoing symptomatic COVID-19, as defined by persisting physical symptoms (respiratory symptoms, fatigue, impaired sleep and number of other symptoms), mental health symptoms and inability to return to work if employed. Descriptive statistics and multiple regression analyses were used to compare differences in characteristics, follow-up outcomes and blood tests between ethnic groups. To account for possible selection bias, analyses were adjusted for propensity scores.

### Results

986 adults completed follow-up: 202 (20.5%) Asian, 105 (10.6%) black, 18 (1.8%) mixed, 468 (47.5%) white and 111 (11.3%) from other backgrounds. Differences between groups included white adults being older than those from Asian/'other' backgrounds and black adults being more likely from deprived areas than those from Asian/white/'other'

individuals situated within a local area and contains sensitive information about protected characteristics including ethnicity. There would be a high risk of re-identification even with sharing an anonymised data set. Data can be requested from the Research and Development department at rf-tr.randd@nhs.net. Data can be cited as: Emmanuel Way, "Extending the electronic Clinical Infection Database (eLCID) tool to support clinicians and researchers in the COVID-19 response", Royal Free Hospital NHS Trust, United Kingdom.

**Funding:** The author(s) received no specific funding for this work.

**Competing interests:** The authors have declared that no competing interests exist.

backgrounds. After adjusting for these differences, at follow-up, black adults had fewer respiratory (adjusted odds ratio 0.49 (0.25–0.96)) and other symptoms (adjusted count ratio 0.68 (0.34–0.99)) compared to white adults. There were otherwise no significant differences between ethnic groups in terms of physical health, mental health or ability to return to work if employed. These findings were not altered after adjustment for propensity scores.

## Conclusions

In our population, despite having more co-morbidities associated with worse outcomes, adults from Asian, black, mixed and other ethnic backgrounds are not more likely to develop ongoing symptomatic COVID-19. However, it is important that healthcare services remain vigilant in ensuring the provision of timely patient-centred care.

## Introduction

As countries adapt to a world with SARS-CoV-2 infection, and effective vaccinations are available to reduce risk of acute infection, adults with 'ongoing symptomatic COVID-19' and 'Long COVID' [1] are an increasing focus of attention. In the United Kingdom, individuals with signs and symptoms persisting four to 12 weeks after their acute illness are defined as having 'ongoing symptomatic COVID-19'; the majority will continue to experience symptoms after 12 weeks or 'Long COVID' [2]. Approximately 1.8 million adults (3.5% of the population) in the UK have reported having Long COVID [3]. This has economic and social consequences individuals and healthcare systems [4].

Data from the UK and USA show adults from certain ethnic groups such as Asian, black and Hispanic have a greater chance of acute COVID-19 infection than white adults and may be more likely to be hospitalised and die [5, 6]. Worse outcomes may persist immediately after discharge with an increased risk of death, readmission and multiorgan dysfunction [7]. Several biochemical and socioeconomic mechanisms have been proposed to explain this, such as differences in host response to COVID-19, increased comorbidity, poorer quality or differing access to healthcare and different working and living conditions [8, 9]. Structural and institutional racism, where systems and policies discriminate against and disadvantage minoritised ethnic groups, compounds these factors, such as by generating and reinforcing inequities in access to resources and exposure to COVID-19 and therefore drives inequalities in health and socioeconomic outcomes [10].

There are less data about the relationship between ethnicity and ongoing symptomatic COVID-19. Studies are also limited by aggregating ethnic groups into 'white' and 'non-white' or 'black, Asian and minority ethnic' (BAME) [11]. Ethnic groups aggregated into 'non-white' differ in terms of factors such as deprivation and comorbidities; for example, South Asian adults are more likely to have ischaemic heart disease than white adults while black adults have more hypertension but lower ischaemic heart disease than white adults [11]. Aggregating ethnic groups therefore masks differences between groups and limits understanding of heterogenous experiences across groups.

It is clearly important to prioritise ethnic minorities who have been disproportionately affected by acute SARS-CoV-2 infection in research on ongoing symptomatic COVID-19. Our objective was to assess if, among a population followed up after discharge from hospital with

COVID-19, adults from Asian, black, mixed and other backgrounds are at increased risk of physical and mental health symptoms.

## Methods

### Study design and participants

In Spring 2020, we established a virtual review (methodology and questionnaire previously reported [12]) for all adults diagnosed with COVID-19 (defined clinically or tested positive within 7 days of admission) six to 12 weeks post-discharge. Effort was made to follow up all patients, including by using translation services for non-English speakers. Our cohort included adults aged 18 and above admitted to both inpatient wards and intensive care (ICU) from 03/03/2020 to 27/11/2021. This marked the period from the beginning of the first wave in our hospital to before the Omicron variant started to dominate in the UK [13]. Adults who acquired COVID-19 in hospital, died within six weeks of discharge, were too frail to complete follow-up or were followed-up elsewhere were excluded from follow-up.

Our cohort was aggregated into five ethnic groups in line with the UK government's 2021 census [14]. These were (in alphabetical order): Asian (including Asian British, Bangladeshi, Chinese, Indian and Pakistani), black (including black British, African and Caribbean), mixed, white (including British Irish, English, Gypsy or Irish Traveller, Irish, Northern Irish, Roma Scottish, Welsh and White British) and other (including Arab). Individuals were grouped based on self-reported ethnicity if available or by electronic health records. Those with unrecorded ethnicities were excluded from analysis.

Data were collected on demographics, comorbidities and admission details as well as symptom burden and blood tests at follow-up. We consecutively sampled and analysed all eligible adults who had been discharged and consented to follow up. Ethical approval was provided by Health Research Authority and Health and Care Research Wales (HRA number 20/HRA/492). SBN, AS, SB, KT, CNN, GS, HMB were involved in service delivery and data collection of identifiable information. Data were accessed for this project on 9[th] December 2022 and were pseudo-anonymised prior to analysis.

### Statistical analysis

All variables were tested for parametric assumptions prior to performing statistical tests. Parametric assumptions were not met due to non-normality of numeric variables and small sample size in the mixed group. Continuous data were presented as medians with interquartile range (IQR) and categorical data as proportions. Continuous data were compared using Kruskal-Wallis test and categorical data with Fisher's exact test and post-hoc analysis was performed with Bonferroni correction for these comparisons to determine differences within groups.

Multiple regression analyses were performed to compare follow-up outcomes between ethnic groups without adjustment, adjusting for age and sex only and adjusting for clinically and statistically relevant factors (using Bayesian information criterion (BIC) and bivariate associations) [15]. Outcomes measured at follow-up were presence of respiratory symptoms (cough and/or shortness of breath), fatigue, poor sleep quality, number of other symptoms (out of 10: chest pain, chest tightness, myalgia, anosmia, anorexia, abdominal pain, diarrhoea, leg swelling, confusion and focal weakness), affected mental health (as quantified by either a Patient Health Questionnaire 2-item to screen for depression or Trauma Screening Questionnaire (TSQ) for post-traumatic stress disorder) and inability to return to work if employed. Negative binomial regression was used for number of symptoms, and binary logistic regression for all other follow-up outcomes (which were binary). Data presented as count ratios (CR) for

negative binomial regression and odds ratios (OR) for logistic regression. To account for possible selection bias, regression analyses adjusted for propensity scores were also performed.

Co-variates included age, index of deprivation, body mass index (BMI), clinical frailty score (CFS) and time to follow-up as continuous variables, and sex, hypertension, diabetes, smoking status (ever- and never-smokers), wave, severity of acute COVID-19 and inpatient treatment with steroids as categorical variables. Waves were wild-type (03/03/2020–07/09/2020), alpha (08/09/2020–24/04/2021) and delta (25/04/2021–27/11/2021). Severity of acute COVID-19 was determined in accordance with World Health Organisation clinical progression scale as moderate (hospitalised with no oxygen or oxygen via mask and/or nasal prongs) or severe (hospitalised with oxygen via non-invasive ventilation, high-flow nasal oxygen and/or invasive ventilation) [16]. Blood tests (haemoglobin, fibrinogen, D-dimer, C-reactive protein, ferritin, N-terminal pro b-type natriuretic peptide (NT-proBNP) and troponin T) at follow-up were adjusted for admission results. Linear regression was used for blood tests.

All tests of significance were two-tailed and p-values ≤0.05 were considered statistically significant. Post-hoc analyses of continuous data using Kruskal-Wallis test and categorical data using Fisher's exact test were adjusted using the Bonferroni correction. All statistical analyses were performed in SPSS (Version 27.0).

## Results

We discharged 1456 adults who were originally admitted between 03/03/2020 to 27/11/2021. Follow-up was completed at a median of 55 days (IQR 46–79) in 986 (67.7%) adults: 202 (20.5%) Asian, 105 (10.6%) black, 18 (1.8%) mixed, 468 (47.5%) white and 111 (11.3%) from other ethnic backgrounds. Ethnicity information was derived from self-report for 651/986 (66.0%) of adults and from electronic health records where self-report was unavailable. Eighty-four adults had unrecorded ethnicities and were excluded from analysis. Most demographic factors and co-morbidities and all key outcomes at follow-up did not differ significantly between those whose ethnicities were known and unknown (S1 Table).

### Demographics, co-morbidities and admission details

Our data includes a small number of individuals (34, 3.45%) who were clinically diagnosed with COVID-19 and either had a negative swab (n = 16) or a positive swab more than seven days from admission. All other individuals had a positive swab within seven days of admission.

There were significant differences in demographic factors and co-morbidities between ethnic groups (Table 1). Post-hoc analyses adjusted using the Bonferroni correction showed that adults from white backgrounds were older than those from Asian or 'other' backgrounds (median age 64.0 vs. 59.5 and 57.0, p<0.001) and more likely than Asian adults to have received at least one vaccine dose prior to admission if eligible (21% vs. 4%, p = 0.02). Black adults were more likely to be from deprived areas than adults from Asian, white or 'other' backgrounds (median deprivation decile 3 vs. 5, 6 and 5, p<0.001), have higher body mass index (BMI) than Asian or white adults (median 28.95kg/m$^2$ vs. 25.77kg/m$^2$ and 26.99 kg/m$^2$, p = 0.006) and have hypertension compared to adults from white or 'other' backgrounds (58% vs. 42% and 39%, p = 0.02). Adults from Asian backgrounds were less likely to have ever smoked than those from white or 'other' backgrounds (22% vs. 40% and 38%, p≤0.04). Both Asian and black adults were more likely to have diabetes than white adults (32% and 31% vs. 15%, p<0.001). There were no significant differences in sex or in other co-morbidities.

Adults across all ethnic groups had a similar number of symptoms at presentation, vital signs as measured by National Early Warning Score (NEWS2), and severity of infection. Adults from Asian and white backgrounds were more likely than those from 'other'

**Table 1. Demographics, co-morbidities, admission and follow-up characteristics of adults from different ethnic groups.**

| Aggregated ethnic group | Asian (n = 200) | Black (n = 105) | Mixed (n = 18) | White (n = 468) | Other (n = 111) | p-value |
|---|---|---|---|---|---|---|
| **Demographics and co-morbidities** | | | | | | |
| Age (years) | 59.50[d] (47.25–70.00) | 60.00 (51.00–69.00) | 55.00 (50.50–69.50) | 64.00[a,e] (52.00–76.00) | 57.00[d] (49.00–65.00) | **<0.001** |
| Sex (female) (%) | 86 (43) | 48 (46) | 5 (28) | 185 (40) | 40 (36) | 0.42 |
| Deprivation decile* | 5.00[b] (3.00–7.00) | 3.00[a,d,e] (2.00–5.00) (n = 103) | 5.50 (3.00–7.00) | 6.00[b] (4.00–7.00) | 5.00[b] (3.00–7.00) | **<0.001** |
| Body mass index (kg/m$^2$) | 25.77[b] (23.66–29.81) | 28.95[a,d] (25.66–33.75) | 27.68 (23.51–31.63) | 26.99[b] (23.53–31.16) | 26.36 (24.17–30.07) | **0.006** |
| Hypertension (%) | 94 (47) | 61 (58)[d,e] | 6 (33) | 195 (42) | 43 (39) | **0.02** |
| Diabetes (%) | 61 (31)[d] | 34 (32)[d] | 2 (11) | 69 (15)[a,b] | 19 (17) | **<0.001** |
| Any cardiac disease (%) | 44 (22) | 21 (20) | 3 (17) | 115 (25) | 23 (21) | 0.78 |
| Cerebrovascular disease (%) | 11 (6) | 12 (11) | 1 (6) | 37 (8) | 11 (10) | 0.37 |
| Chronic lung disease (%) | 33 (17) | 13 (12) | 1 (6) | 95 (20) | 20 (18) | 0.22 |
| Ever smoked (%) | 42/192 (22)[d,e] | 26/102 (26) | 6 (33) | 179/444 (40)[a] | 40/106 (38)[a] | **<0.001** |
| Chronic kidney disease (%) | 31 (16) | 21 (20) | 3 (17) | 51 (11) | 10 (9) | 0.05 |
| Immunosuppressed due to cancer or autoimmune disease (%) | 24 (12) | 19 (18) | 2 (11) | 78 (17) | 11 (10) | 0.23 |
| Mental health condition (%) | 23 (12) | 14 (13) | 2 (11) | 76 (16) | 13 (12) | 0.52 |
| Clinical frailty score | 3.00 (2.00–3.75) | 3.00[e] (2.00–5.00) | 2.00 (1.75–3.00) | 3.00[e] (2.00–4.00) | 3.00[b,d] (2.00–3.00) | **<0.001** |
| **COVID-19-related characteristics** | | | | | | |
| Wave: wild-type | 35 (18)[e] | 28 (27) | 8 (44) | 100 (21)[e] | 42 (38)[a,d] | **<0.001** |
| Wave: alpha | 153 (77)[b,c,e] | 64 (61)[a] | 6 (33)[a,d] | 322 (69)[c] | 64 (58)[a] | |
| Wave: delta (%) | 12 (6) | 13 (12) | 4 (22) | 46 (10) | 5 (5) | |
| Vaccinated on admission (%) | 2/54 (4)[d] | 3/35 (9) | 0/5 (0) | 28/135 (21)[a] | 1/25 (4) | **0.008** |
| Admission number of symptoms[†] | 3.00 (2.00–4.00) | 3.00 (2.00–4.00) | 3.00 (2.75–5.00) | 3.00 (2.00–4.00) | 3.00 (2.00–4.00) | 0.30 |
| National Earning Warning System 2 Score (NEWS2) | 4.00 (2.00–6.00) | 3.5 (2.00–6.00) | 6.00 (5.00–7.00) | 4.00 (2.00–6.00) | 4.00 (2.00–6.00) | 0.30 |
| Full escalation (%) | 166/178 (93) | 85/93 (91) | 15/15 (100) | 375/417 (90) | 97/105 (92) | 0.61 |
| Severe infection (%) | 41/176 (23) | 18/92 (20) | 1/14 (7) | 90/432 (21) | 18/100 (18) | 0.65 |
| Steroids prescribed (%) | 105/136 (77)[e] | 57/68 (69) | 7/14 (50) | 229/318 (72)[e] | 45/81 (56)[a,d] | **0.006** |
| Other drugs e.g. tocilizumab, remdesivir prescribed (%) | 40/114 (35) | 20/71 (28) | 6/15 (40) | 103/288 (36) | 18/77 (23) | 0.23 |
| Time to follow-up (days) | 55.50 (45.00–78.50) | 59.00 (48.00–86.00) | 64.00 (54.25–91.50) | 54.00 (45.00–77.00) | 58.00 (49.00–84.00) | **0.03** |
| **Follow-up outcomes** | | | | | | |
| Respiratory symptoms** | 101/195 (51.8%) | 45/105 (42.9%) | 11/18 (61.1%) | 246/455 (54.1%) | 57/107 (53.3%) | 0.29 |
| Fatigue | 122/194[a,b] (62.9%) | 56/105[b] (53.3%) | 16/18[a] (88.9%) | 276/454[a,b] (60.8%) | 58/107[a,b] (54.2%) | **0.03** |
| Poor sleep | 68/191 (35.6%) | 32/105 (30.5%) | 8/18 (44.4%) | 166/445 (37.3%) | 41/107 (38.3%) | 0.65 |
| Number of symptoms at follow-up[††] | 0.00 (0.00–1.00) N = 200 | 0.00 (0.00–1.00) N = 105 | 0.00 (0.00–2.00) N = 18 | 0.00 (0.00–1.00) N = 467 | 0.00 (0.00–1.00) N = 111 | 0.24 |
| Affected mental health | 23/200 (11.5%) | 12/105 (11.4%) | 2/18 (11.1%) | 61/468 (13.0%) | 14/111 (12.6%) | 0.98 |
| Inability to return to work | 40/196 (20.4%) | 17/102 (16.7%) | 2/18 (11.1%) | 99/450 (22.0%) | 21/105 (20.0%) | 0.79 |

Data are presented as median (interquartile range) for non-parametric data and proportions (in percentages) for categorical data. Indicates significant adjusted post-hoc Bonferroni compared to adults from (a) Asian (b) black (c) mixed (d) white and (e) other backgrounds.

*Lower deciles indicate higher deprivation.

[†]Number of symptoms out of 19 and included: cough, shortness of breath, sore throat, rhinitis, fever, chills, fatigue, myalgia, headache, anorexia, anosmia, loss of taste, diarrhoea, abdominal pain, chest pain, chest tightness, confusion, peripheral oedema and focal weakness.

**Presence of cough and/or shortness of breath

[††]Number of symptoms out of 10 and included: chest pain, chest tightness, myalgia, anosmia, anorexia, abdominal pain, diarrhoea, leg swelling, confusion and focal weakness

backgrounds to receive steroids during their admission (77% and 72% *vs*. 56%, p<0.05); there was no difference in other drugs prescribed including anti-IL6 antibodies such as tocilizumab or antivirals such as remdesivir.

## Follow-up symptoms

At follow-up, adults continued to experience symptoms consistent with ongoing symptomatic COVID-19 including cough and/or shortness of breath (52.3%), fatigue (60.1%) poor sleep (36.4%) and affected mental health (14.3%).

When adjusted for age and sex only, black adults were less likely to have respiratory symptoms (adjusted odds ratio (aOR) 0.65 (0.42–1.00), p = 0.05) and fewer other symptoms (adjusted count ratio (aCR) 0.61 (0.43–0.87), p = 0.006) while adults from mixed backgrounds were more likely to be fatigued (aOR 5.44 (1.23–24.00), p = 0.03) than white adults at follow-up (see Table 2 for unadjusted and adjusted ratios).

When adjusting for covariates including comorbidities and socioeconomic position, black adults continued to be less likely to have respiratory symptoms (aOR 0.49 (0.25–0.96), p = 0.04) and fewer other symptoms (aCR 0.68 (0.34–0.99), p = 0.046). However, adults from mixed backgrounds were no longer more likely to be fatigued (aOR 3.79 (0.44–33.01), p = 0.23). There were otherwise no significant differences between ethnic groups in terms of physical health symptoms, mental health burden or ability to return to work if employed. These findings were not altered after adjustment for propensity scores to account for selection bias (S2 Table).

For all ethnic groups, factors which were correlated with respiratory symptoms were: never-smoking status (OR 0.63 (0.40–0.99), p = 0.04) and moderate severity of acute COVID-19 (0.56 (0.33–0.95), p = 0.03) (see S3 Table for covariates). Presentation during alpha wave was correlated with number of other symptoms (CR 4.13 (1.47–11.60), p = 0.01), fatigue (OR 2.71 (1.06–6.90), p = 0.04) and poor sleep quality (OR 11.85 (1.50–93.16, p = 0.02) when compared to presentation during delta wave. Lack of hypertension was also correlated with sleep quality (OR 0.63 (0.40–0.99), p = 0.05).

## Follow-up blood tests

There were differences between ethnic groups in blood tests at follow-up associated with ongoing symptomatic COVID-19 (Table 3). For example, Asian and black adults had lower haemoglobin levels and adults from 'other' backgrounds had lower ferritin levels than white adults, however, this difference was attenuated when adjusting for covariates including COVID-19 severity.

Black adults had higher D-dimer results than white adults, even after adjusting for covariates (528ng/mL ((258–798) higher than white adults, p<0.001).

## Discussion

Adults discharged from hospital after COVID-19 are likely to continue suffering from symptoms. The risk of ongoing symptomatic COVID-19 may be increased by certain demographic factors and pre-existing health conditions such as older age [17] and hypertension [18]. Our groups did differ in some of these characteristics; black adults, for example, had increased comorbidities which are usually associated with poorer recovery from COVID-19. However, our data demonstrate that after adjusting for these differences, adults discharged from hospital from Asian, black, mixed and other ethnic backgrounds are not more likely to have ongoing symptoms associated with COVID-19. In fact, at follow-up, black adults were likely to have fewer respiratory and 'other' symptoms at follow-up.

**Table 2. Follow-up outcomes of adults from different ethnic groups.**

|  | Unadjusted OR (95% CI) | p-value | Age- and sex-adjusted OR (95% CI) | p-value | Adjusted OR (95% CI) [‡] | p-value |
|---|---|---|---|---|---|---|
| **Respiratory symptoms**[*] |  |  |  |  |  |  |
| Asian | 0.91 (0.65–1.28) | 0.60 | 0.94 (0.67–1.32) | 0.70 | 0.62 (0.37–1.06) | 0.08 |
| Black | **0.64 (0.42–0.98)** | **0.04** | **0.65 (0.42–1.00)** | **0.05** | **0.49 (0.25–0.96)** | **0.04** |
| Mixed | 1.34 (0.51–3.51) | 0.56 | 1.35 (0.51–3.56) | 0.54 | 0.51 (0.10–2.47) | 0.40 |
| White | 1.00 (ref) | - | 1.00 (ref) | - | 1.00 (ref) | - |
| Other | 0.97 (0.64–1.48) | 0.88 | 0.99 (0.65–1.51) | 0.95 | 1.05 (0.57–1.93) | 0.87 |
| **Fatigue** |  |  |  |  |  |  |
| Asian | 1.09 (0.77–1.55) | 0.62 | 1.13 (0.80–1.60) | 0.50 | 0.95 (0.55–1.64) | 0.86 |
| Black | 0.74 (0.48–1.13) | 0.16 | 0.75 (0.49–1.16) | 0.20 | 0.63 (0.32–1.24) | 0.18 |
| Mixed | **5.16 (1.17–22.7)** | **0.03** | **5.44 (1.23–24.00)** | **0.03** | 3.79 (0.44–33.01) | 0.23 |
| White | 1.00 (ref) | - | 1.00 (ref) | - | 1.00 (ref) | - |
| Other | 0.76 (0.50–1.17) | 0.21 | 0.80 (0.52–1.22) | 0.30 | 0.76 (0.42–1.38) | 0.37 |
| **Poor sleep quality** |  |  |  |  |  |  |
| Asian | 0.93 (0.65–1.32) | 0.68 | 0.93 (0.65–1.32) | 0.68 | 0.74 (0.43–1.28) | 0.28 |
| Black | 0.74 (0.47–1.17) | 0.19 | 0.74 (0.47–1.17) | 0.19 | 0.56 (0.27–1.13) | 0.10 |
| Mixed | 1.35 (0.52–3.47) | 0.54 | 1.33 (0.52–3.45) | 0.55 | 0.95 (0.18–4.97) | 0.95 |
| White | 1.00 (ref) | - | 1.00 (ref) | - | 1.00 (ref) | - |
| Other | 1.04 (0.68–1.61) | 0.85 | 1.04 (0.67–1.61) | 0.87 | 1.21 (0.67–2.19) | 0.53 |
| **Number of symptoms at follow-up**[†] |  |  |  |  |  |  |
| Asian | 0.75 (0.52–1.08) | 0.12 | 1.02 (0.80–1.30) | 0.90 | 0.74 (0.50–1.10) | 0.14 |
| Black | 0.67 (0.41–1.10) | 0.11 | **0.61 (0.43–0.87)** | **0.006** | **0.68 (0.34–0.99)** | **0.046** |
| Mixed | 1.18 (0.40–3.40) | 0.77 | 1.09 (0.55–2.16) | 0.81 | 1.26 (0.42–3.80) | 0.68 |
| White | 1.00 (ref) | - | 1.00 (ref) | - | 1.00 (ref) | - |
| Other | 0.79 (0.52–1.22) | 0.29 | 0.97 (0.71–1.32) | 0.84 | 0.75 (0.48–1.19) | 0.22 |
| **Affected mental health** |  |  |  |  |  |  |
| Asian | 0.87 (0.52–1.45) | 0.58 | 0.89 (0.53–1.50) | 0.67 | 0.84 (0.38–1.84) | 0.66 |
| Black | 0.86 (0.45–1.66) | 0.67 | 0.89 (0.46–1.72) | 0.72 | 1.44 (0.56–3.70) | 0.45 |
| Mixed | 0.83 (0.19–3.72) | 0.81 | 0.84 (0.19–3.77) | 0.82 | 2.98 (0.46–19.28) | 0.25 |
| White | 1.00 (ref) | - | - | - | 1.00 (ref) | - |
| Other | 0.96 (0.52–1.79) | 0.91 | 0.99 (0.53–1.84) | 0.96 | 0.78 (0.30–2.05) | 0.61 |
| **Inability to return to work** |  |  |  |  |  |  |
| Asian | 1.05 (0.65–1.70) | 0.83 | 1.03 (0.63–1.67) | 0.91 | 1.16 (0.55–2.45) | 0.69 |
| Black | 1.22 (0.63–2.35) | 0.56 | 1.20 (0.62–2.31) | 0.59 | 0.62 (0.22–1.70) | 0.35 |
| Mixed | 1.85 (0.35–9.72) | 0.47 | 1.82 (0.34–9.60) | 0.48 | 0.69 (0.09–5.49) | 0.73 |
| White | 1.00 (ref) | - | 1.00 (ref) | - | 1.00 (ref) | - |
| Other | 0.92 (0.49–1.72) | 0.78 | 0.91 (0.48–1.71) | 0.76 | 0.91 (0.38–2.17) | 0.82 |

[‡]Adjusted for age, sex, index of deprivation, BMI, hypertension, diabetes, smoking status, clinical frailty score, wave, treatment with inpatient steroids, maximum respiratory support received and time (in days) to follow-up. Count ratios are presented for number of symptoms at follow-up.

[*]Presence of cough and/or shortness of breath.

[†]Number of symptoms out of 10 and included: chest pain, chest tightness, myalgia, anosmia, anorexia, abdominal pain, diarrhoea, leg swelling, confusion and focal weakness

Ethnic groups did differ in terms of persisting biochemical abnormalities. Of note, black adults had greater derangement of D-dimer, even after adjusting for covariates. Ongoing deranged blood tests may reflect the risk of organ dysfunction or may correlate with persistent symptoms; increased D-dimer may, for example, predict pulmonary dysfunction [19].

**Table 3. Linear regression of follow-up blood tests of adults from different ethnic groups.**

| | Unadjusted (95%) CI | p-value | Age- and sex-adjusted (95% CI)[&] | p-value | Adjusted (95% CI) [‡] | p-value |
|---|---|---|---|---|---|---|
| **Haemoglobin (g/L)** | | | | | | |
| Asian | **-5.3 (-8.6 –-2.0)** | **0.002** | **-2.3 (-4.5–0.00)** | **0.05** | -1.5 (-4.7–1. 8) | 0.38 |
| Black | **-10.7 (-15.2 –-6.1)** | **<0.001** | **-4.6 (-7.8 - -1.5)** | **0.004** | -2.3 (-6.8–2.3) | 0.33 |
| Mixed | 2.7 (-8.1–13.4) | 0.63 | -3.0 (-10.5–4.6) | 0.44 | -1.9 (-14.8–11.1) | 0.78 |
| White | 1.0 (ref) | - | 1.0 (ref) | - | 1.0 (ref) | - |
| Other | 1.6 (-3.0–6.2) | 0.49 | 1.7 (-1.4–4.8) | 0.29 | 0.6 (-3.7–4.9) | 0.79 |
| **Fibrinogen (g/dL)** | | | | | | |
| Asian | -0.1 (-0.3–0.1) | 0.41 | 0.0 (-0.2–0.2) | 0.71 | -0.2 (-0.4–0.1) | 0.14 |
| Black | -0.0 (-0.3–0.2) | 0.90 | 0.1 (-0.2–0.4) | 0.50 | -0.2 (-0.5–0.2) | 0.33 |
| Mixed | -0.4 (1.0–0.2) | 0.18 | -0.2 (-0.8–0.4) | 0.50 | -0.5 (-1.2–0.2) | 0.15 |
| White | 1.0 (ref) | - | 1.0 (ref) | - | 1.0 (ref) | - |
| Other | -0.2 (-0.4–0.1) | 0.18 | 0.0 (-0.3–0.2) | 0.94 | -0.1 (-0.4–0.1) | 0.37 |
| **D-dimer (ng/mL)** | | | | | | |
| Asian | -110 (-357–137) | 0.38 | -24 (-286–237) | 0.85 | 59 (-138–255) | 0.56 |
| Black | **656 (319–992)** | **<0.001** | **637 (283–991)** | **<0.001** | **528 (258–798)** | **<0.001** |
| Mixed | 91 (-659–841) | 0.81 | 317 (-506–1140) | 0.45 | **759 (190–1327)** | **0.009** |
| White | 1 (ref) | - | 1 (ref) | - | 1 (ref) | - |
| Other | -39 (-343–265) | 0.80 | 120 (-196–435) | 0.46 | 59 (-169–287) | 0.61 |
| **Albumin (g/L)** | | | | | | |
| Asian | 0 (-1–1) | 0.91 | 0 (-1–1) | 0.45 | -1 (-3–1) | 0.22 |
| Black | -1 (-3–0) | 0.10 | -1 (-2–1) | 0.26 | -2 (-5–0) | 0.05 |
| Mixed | 1 (-2–6) | 0.34 | 1 (-2–5) | 0.52 | 1 (-7–8) | 0.86 |
| White | 1 (ref) | - | 1 (ref) | - | 1. (ref) | - |
| Other | 0 (-2–1) | 0.80 | -1 (-2–1) | 0.46 | -1 (-3–1) | 0.47 |
| **C-reactive protein (mg/mL)** | | | | | | |
| Asian | 0 (-5–4) | 0.89 | 1 (-4–5) | 0.79 | -1 (-6–4) | 0.74 |
| Black | -2 (-8–3) | 0.46 | -2 (-7–4) | 0.55 | -3 (-10–3) | 0.29 |
| Mixed | -7 (-20–5) | 0.24 | -6 (-19–7) | 0.34 | -3 (-18–11) | 0.67 |
| White | 1 (ref) | - | 1 (ref) | - | 1 (ref) | - |
| Other | -4 (-9–2) | 0.20 | -2 (-8–3) | 0.39 | -3 (-9–3) | 0.33 |
| **Ferritin (ug/L)** | | | | | | |
| Asian | -63 (-160–33) | 0.20 | -72 (-168–25) | 0.15 | 16 (-112–143) | 0.81 |
| Black | -69 (-203–66) | 0.32 | -102 (-241–36) | 0.15 | -71 (-252–109) | 0.44 |
| Mixed | 97 (-203–397) | 0.53 | -63 (-367–241) | 0.69 | -210 (-641–221) | 0.34 |
| White | 1 (ref) | - | 1 (ref) | - | 1 (ref) | - |
| Other | **-153 (-275 - -32)** | **0.01** | **-143 (-264 –-22)** | **0.02** | -101 (-248–47) | 0.18 |
| **N-terminal pro b-type natriuretic peptide** (ng/L) | | | | | | |
| Asian | 1138 (11–2265) | 0.05 | 283 (-440-1007) | 0.443 | -335 (-1053–384) | 0.36 |
| Black | 885 (-651–2422) | 0.26 | 630 (-382–1642) | 0.223 | **1623 (601–2645)** | **0.002** |
| Mixed | **4520 (1074–7966)** | **0.01** | 1990 (-224–4203) | 0.08 | **5176 (2786–7567)** | **<0.001** |
| White | 1 (ref) | - | 1 (ref) | - | 1 (ref) | - |
| Other | -529 (-1932–874) | 0.46 | -129 (-1037–779) | 0.78 | 22 (-811–855) | 0.96 |
| **Troponin T (ng/L)** | | | | | | |
| Asian | -1.8 (-16.7–13.1) | 0.81 | -0.3 (-17.4–16.8) | 0.97 | -3.2 (-35.8–29.4) | 0.85 |
| Black | 0.4 (-20.0–20.8) | 0.97 | 2.3 (-22.5–27.1) | 0.86 | 12.0 (-36.1–60.2) | 0.62 |
| Mixed | 2.5 (-42.0–47.0) | 0.91 | 5.7 (-44.7–56.1) | 0.83 | 27.1 (-68.4–122.6) | 0.58 |
| White | 1.0 (ref) | - | 1.0 (ref) | - | 1.0 (ref) | - |

(*Continued*)

**Table 3.** (Continued)

|  | Unadjusted (95%) CI | p-value | Age- and sex-adjusted (95% CI)[&] | p-value | Adjusted (95% CI) [‡] | p-value |
|---|---|---|---|---|---|---|
| Other | -12.7 (-31.5–6.2) | 0.19 | -10.4 (-32.0–11.2) | 0.35 | -10.1 (-48.7–28.6) | 0.61 |

[&]Adjusted for age, sex and corresponding admission blood test.

[‡]Adjusted for age, sex, index of deprivation, BMI, hypertension, diabetes, smoking status, clinical frailty score, wave, treatment with inpatient steroids, maximum respiratory support received, time (in days) to follow-up and corresponding admission blood test.

However, our data do not demonstrate a clear relationship between symptoms and blood tests. The clinical significance of such biochemical abnormalities in individuals is uncertain and requires further investigation.

There is some evidence that earlier waves, such as wild-type and alpha, are associated with increased likelihood of persistent post-COVID symptoms [20] and our data are consistent with this. In this report we have not discussed the Omicron variant. It is possible that certain ethnic groups, such as black adults, are more likely to be hospitalised with the Omicron variant [21]. However, Asian and black adults were already known to be at increased risk of infection and hospitalisation in previous waves [5, 6], and our data showed no increased risk of ongoing symptomatic COVID-19 associated with these variants. Adults with the Omicron variant may also be less likely to experience persistent COVID symptoms [20] and may therefore require less healthcare resource allocation. Furthermore, research shows there remains a high burden of symptoms among survivors of acute COVID-19 infection even at two years [22]. It therefore remains important to understand the factors contributing to persistent symptoms post COVID-19 in those infected in earlier waves.

Our data includes a small number of individuals (3.45%) who were clinically diagnosed with COVID-19. This rate compares favourably, for example to the RECOVERY trial which included participants with clinically suspected or laboratory-confirmed SARS-CoV-2 infection and where approximately 10% of participants had a negative SARS-CoV-2 test result [23].

Data about the impact of ethnicity on symptoms post-COVID-19 has been inconsistent with some studies suggesting increased risk of symptoms such as breathlessness among black adults and others suggesting there are fewer symptoms overall among Asian and black adults [24–27]. However, these studies are highly heterogenous and thus any meta-analyses have been limited [27] and may not be appropriate. For example, studies may not follow up non-English-speaking participants [24] or may have Asian and black adults particularly under-represented [26]. Furthermore, adults from ethnic minorities have been shown to have reduced access and engagement with healthcare in the UK [28, 29]; relying on individuals to respond to surveys or access healthcare to be diagnosed may thus disproportionately exclude these individuals. Our study has a larger proportion of adults from Asian, black, mixed and other backgrounds than most reported studies. This may be because of pro-active efforts to follow up all patients, including using translation services, resulting in a cohort which is representative of the multi-cultural population that our hospital serves [30].

It may be surprising that ethnicity has been associated with increased infection risk and mortality among some ethnic groups [5, 6], but not with ongoing symptomatic COVID-19 in our cohort. One reason may be that the severity of acute infection has a greater role in increasing the risk of developing persistent symptoms [15, 17]. Unlike other studies [5], our data show no difference in severity of infection between ethnic groups among adults who survived acute SARS-CoV-2 infection. The biomedical factors and social determinants of health which may contribute to increased infection risk and mortality from acute SARS-CoV-2 infection among some ethnic groups [8] could also mitigate the risk of ongoing symptomatic COVID-

19. For example, data have shown that black adults were more likely to be seropositive after SARS-CoV-2 infection [31] and stronger antibody response may be associated with both increased severity of acute SARS-CoV-2 infection and reduced risk of ongoing symptomatic COVID-19 [32, 33]. Increased social interaction both personally and professionally may increase risk of infection among some ethnic groups [8], but this increased social capital may also improve resilience and therefore recovery from COVID [34]. Viewing post-COVID-19 symptoms through the lens of a socio-ecological model, where different interrelated factors impact health, may help explain its predictors [35].

Our study is limited as it aggregates ethnicities into five main groups; for example, adults from Indian, Bangladeshi and Pakistani backgrounds are grouped together. Combining these different groups may mask differences, for example, adults from Bangladeshi and Pakistani backgrounds have a higher age-standardised mortality rate from SARS-CoV-2 infection than those from Indian backgrounds [36]. Our sample also comprises of a small number of adults from mixed backgrounds and limits conclusions about this group; for example, confidence intervals for follow-up outcomes such as fatigue were large (0.44–33.01). Larger and more representative analyses of ongoing symptomatic COVID-19 and Long COVID by ethnicity are therefore required. However, ensuring that ethnicity is documented accurately is a potential challenge in a larger cohort, especially for adults from mixed groups [37]. Our data are also limited as approximately a third of participants' ethnicities were determined from electronic health records, which may misclassify participants' ethnicities. We also present the experience of a single follow-up centre in the UK and data may not be generalisable to all populations. Finally, our data are limited by the relatively shorter follow-up time; nonetheless there appears to be a good correlation between symptoms reported at two months and those reported at one year [2].

In conclusion, follow-up data from 986 adults discharged post COVID-19 hospitalisation demonstrates that in our population, unlike acute COVID-19 infection, adults from Asian, black, mixed and other ethnic backgrounds are not more likely than white adults to experience ongoing symptoms from COVID-19. However, given the increased risk of infection, there may be more adults from these ethnic groups within follow-up services. It is important that healthcare services remain vigilant to structural discrimination and ensure the provision of timely patient-centred care.

## Supporting information

**S1 Checklist. Human participants research checklist.**
(DOCX)

**S1 Table. Comparison between observed characteristics of participants with known and unknown ethnicities.**
(DOCX)

**S2 Table. Follow-up outcomes of adults from different ethnic groups adjusted for propensity score.**
(DOCX)

**S3 Table. Variables associated with follow-up outcomes.**
(DOCX)

## Acknowledgments

We acknowledge AS Hunter and D Miller for aiding with data acquisition.

## Author Contributions

**Conceptualization:** Sindhu Bhaarrati Naidu, Anita Saigal, John Robert Hurst, Marc Caeroos Isaac Lipman, Swapna Mandal.

**Data curation:** Sindhu Bhaarrati Naidu, Shiuli Bhattacharyya, Karthig Thillaivasan, Camila Nagoda Niklewicz, George Seligmann, Heba Majed Bintalib.

**Formal analysis:** Sindhu Bhaarrati Naidu, Anita Saigal, Amar Jitu Shah, Chibueze Ogbonnaya, Songyuan Xiao, Swapna Mandal.

**Investigation:** Swapna Mandal.

**Methodology:** Sindhu Bhaarrati Naidu, John Robert Hurst, Marc Caeroos Isaac Lipman, Swapna Mandal.

**Project administration:** Sindhu Bhaarrati Naidu, Shiuli Bhattacharyya, Swapna Mandal.

**Resources:** Swapna Mandal.

**Supervision:** John Robert Hurst, Marc Caeroos Isaac Lipman, Swapna Mandal.

**Validation:** John Robert Hurst, Swapna Mandal.

**Writing – original draft:** Sindhu Bhaarrati Naidu.

**Writing – review & editing:** Sindhu Bhaarrati Naidu, Anita Saigal, Amar Jitu Shah, Chibueze Ogbonnaya, Shiuli Bhattacharyya, Karthig Thillaivasan, Songyuan Xiao, Camila Nagoda Niklewicz, George Seligmann, Heba Majed Bintalib, John Robert Hurst, Marc Caeroos Isaac Lipman, Swapna Mandal.

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
