## [Decision Letter · Decision Letter 0]

29 Aug 2024

PONE-D-24-27497Associations between ethnicity and persistent physical and mental health symptoms experienced as part of ongoing symptomatic COVID-19PLOS ONE

Dear Dr. Naidu,

Thank you for submitting your manuscript to PLOS ONE. After careful consideration, we feel that it has merit but does not fully meet PLOS ONE’s publication criteria as it currently stands. Therefore, we invite you to submit a revised version of the manuscript that addresses the points raised during the review process.

We look forward to receiving your revised manuscript.

Kind regards,

Tai-Heng Chen, M.D., Ph.D.

Academic Editor

PLOS ONE

3. We notice that your supplementary tables are included in the manuscript file. Please remove them and upload them with the file type 'Supporting Information'. Please ensure that each Supporting Information file has a legend listed in the manuscript after the references list.

Reviewers' comments:

Reviewer's Responses to Questions

**Comments to the Author**

1. Is the manuscript technically sound, and do the data support the conclusions?

Reviewer #1: Partly

Reviewer #2: Yes

2. Has the statistical analysis been performed appropriately and rigorously? 

Reviewer #1: Yes

Reviewer #2: Yes

3. Have the authors made all data underlying the findings in their manuscript fully available?

Reviewer #1: Yes

Reviewer #2: No

4. Is the manuscript presented in an intelligible fashion and written in standard English?

Reviewer #1: Yes

Reviewer #2: Yes

5. Review Comments to the Author

Reviewer #1: Thank you for the opportunity to review this manuscript. The authors present a small analysis, comparing persistent physical and mental health symptoms across ethnic groups, 6-12 weeks post-discharge following hospitalisation from COVID-19, finding minimal differences across ethnic groups. Given the small sample size, it is difficult to draw reliable conclusions. I have provided more detailed suggestions below:

- In the second paragraph of the introduction, the authors state that several biochemical and socioeconomic mechanisms have been proposed to explain this, yet list primarily socioeconomic mechanisms. The authors should consider focussing more on the role of structural and institutional racism in contributing to ethnic inequalities in COVID-19 health outcomes (e.g., https://www.thelancet.com/journals/eclinm/article/PIIS2589-5370(23)00537-0/fulltext)

- It may be useful to explain why aggregating ethnic groups in previous studies limits our understanding of heterogenous experiences across groups.

- Please include the proportion of participants with ethnicity data determined through electronic health records rather than self-report, as this may bias results due to misclassification of ethnicity.

- The analyses are over-adjusted, as comorbidities and socioeconomic position (IMD) are potential mediators or drivers of ethnic inequalities in COVID-19 health outcomes. Ethnicity itself is not a cause of poor health outcomes, it is that ethnic minority groups are more likely to have comorbidities and lower socioeconomic position and this increases the risk of poor health outcomes. Though the authors do present unadjusted analyses in Table 2 (not stated in the methods), the authors should consider presenting only age and sex adjusted analyses as the primary analyses, then adjusting for comorbidities and socioeconomic position to determine whether controlling for factors attenuates differences between groups.

- in the discussion, the authors state that they are unclear if the large confidence intervals are due to the small sample size or possible clinical significance, but confidence intervals are directly related to the size of the sample. The small sample size of this study limits the ability to determine reliable conclusions. Therefore, the authors should emphasize the need for larger and more representative studies to truly determine whether there are ethnic inequalities in long-COVID-19 symptoms.

Reviewer #2: Firstly, well done for completing this impressive and important piece of work with no external funding. It was an enjoyable and interesting read, well thought through and conducted.

I have some minor comments:

Page 5, final line, and page 14, line 293: please state the ethics review board by name.

Page 6, lines 114-115: “continuous data were presented as medians with interquartile range (IQR) and categorical data as proportions”; for the ease of the reader, please remind them in the results section about this; e.g.

- Page 8, line 159: median age

- Page 8 line 162/163: median deprivation decile

Page 7, lines 143-144, “All tests of significance were two-tailed and p-values ≤0.05 were considered statistically significant.” Would it be helpful to remind the reader that the Bonferroni correction is used? Did you use that for all testing? If not, please can you make it clearer in the methods where did and didn’t apply the Bonferroni correction?

- Same on page 9, line 180.

Page 7, line 148: space missing: “(66.7%)adults”

Page 8, line 158: “Post-hoc Bonferroni analyses”: this is usually called something like ‘post-hoc analyses adjusted using the Bonferroni correction’, please update the language.

Page 9, line 180/181: suggest insertion of the word ‘overall’ to distinguish these results from the results looking at ethnic groups individually (in the supplementary table), i.e. “Post-hoc Bonferroni analyses showed no differences between ethnic groups in terms of unadjusted follow-up outcomes (Table 1).” To “Post-hoc Bonferroni analyses showed no differences between ethnic groups overall in terms of unadjusted follow-up outcomes (Table 1).”

Page 9, line 193/194: “The only factor 194 which was correlated with number of other symptoms was presentation during alpha wave (alpha CR 4.13 (1.47 – 11.60), p = 0.01 vs. delta).” – In the same supplementary table, alpha wave was correlated with fatigue, and poor sleep quality was correlated with both the absence of hypertension and alpha wave. The reader may be interested in these, and I think they should be mentioned in the text as not all readers look at supplementary tables.

Page 11, line 223: “Our data includes a small number of individuals (3.45%) who were clinically diagnosed with COVID-19. This rate compares favourably, for example to the RECOVERY trial which had a false-negative rate of approximately 10%21.” – please make it clearer in the text the distinction between this 3.45% and the other individuals in the study (e.g. positive test result). Please also make it clearer or update the statement about the RECOVERY trial – were these false-negative results? How was that determined?

Page 12, first paragraph: Great – really interesting and relevant discussion of the wider social context – thank you.

Tables: would you consider adding decimal places to some of the median values? Looking at Table 1,

- Age is presumably reported as median (IQR), with 2 decimal places.

- Deprivation decile: only one of the values has one decimal place – please make this more consistent – maybe with decimal places for all the median values reported, and for the IQR?

- Clinical frailty score: the medians are 3, 3, 2, 3, 3, yet the difference between groups is statistically significant. I think reporting decimal places would be useful for the reader. Additionally only two of the values of the IQRs have decimal places, both reported to two decimal places.

- Please go through the tables to check for this; in particular Supplementary Table 1, Covid-19 characteristics, admission number of symptoms; median 3 compared to a median also 3, and the difference is significant.

I suggest you update the row header in tables from “Severity (moderate)” to “Acute COVID-19 Severity (moderate)” for the ease of the reader.

Supplementary table 3, respiratory symptoms, ‘Other’: the result is non significant. Following the rest of your reporting I wouldn’t expect this result to be in bold – is it bold in error?

In summary, well done; I only have very minor comments.

6. PLOS authors have the option to publish the peer review history of their article (what does this mean?). If published, this will include your full peer review and any attached files.

Reviewer #1: **Yes: **Patricia Irizar

Reviewer #2: No

---

## [Author Response · Author response to Decision Letter 0]

16 Sep 2024

London, 11th September 2024

Dear Dr Tai-Heng Chen,

Re: PONE-D-24-27497 Associations between ethnicity and persistent physical and mental health symptoms experienced as part of ongoing symptomatic COVID-19

We thank you and the reviewers for their comments. We have addressed the comments below in a point-by-point response, indicating where we have made changes to the marked copy of the manuscript. We feel the manuscript has been strengthened by these changes.

We appreciate the opportunity to submit this revision as we believe that your readership would be highly interested in the findings of our paper.

Journal requirements

Reply: Thank you, we have reformatted our manuscript as requested. 

2. We note that you have indicated that there are restrictions to data sharing for this study. 

a. If there are ethical or legal restrictions on sharing a de-identified data set, please explain them in detail (e.g., data contain potentially identifying or sensitive patient information, data are owned by a third-party organization, etc.) and who has imposed them (e.g., a Research Ethics Committee or Institutional Review Board, etc.). Please also provide contact information for a data access committee, ethics committee, or other institutional body to which data requests may be sent. 

b. If there are no restrictions, please upload the minimal anonymized data set necessary to replicate your study findings to a stable, public repository and provide us with the relevant URLs, DOIs, or accession numbers

Reply: There are ethical restrictions in sharing a de-identified data set as our data consists of a small number (<1000) of individuals situated within a local area and contains sensitive information about protected characteristics including ethnicity. There would be a high risk of re-identification even with sharing an anonymised data set. 

This research was conducted with the approval of national ethics board and a local COVID-19 ethics committee. The latter has now disbanded. Therefore, data can be requested from the Research and Development department. We have updated our data availability statement accordingly. 

3. We notice that your supplementary tables are included in the manuscript file. Please remove them and upload them with the file type 'Supporting Information'. Please ensure that each Supporting Information file has a legend listed in the manuscript after the references list.

Reply: We have removed the supplementary tables from the manuscript file and included, as per the formatting guide, required information after references. 

Reviewer #1

1. In the second paragraph of the introduction, the authors state that several biochemical and socioeconomic mechanisms have been proposed to explain this, yet list primarily socioeconomic mechanisms. The authors should consider focussing more on the role of structural and institutional racism in contributing to ethnic inequalities in COVID-19 health outcomes (e.g., https://www.thelancet.com/journals/eclinm/article/PIIS2589-5370(23)00537-0/fulltext)

Reply: Thank you for this important point. We agree on the importance of structural and institutional racism. We have updated our introduction to reference an additional biochemical mechanism (line 72). We have also added the recommended reference and emphasised the importance of structural and institutional racism in contributing to other factors (lines 74-75).

2. It may be useful to explain why aggregating ethnic groups in previous studies limits our understanding of heterogenous experiences across groups.

Reply: Thank you, we have updated our introduction as recommended (lines 78-81). 

3. Please include the proportion of participants with ethnicity data determined through electronic health records rather than self-report, as this may bias results due to misclassification of ethnicity.

Reply: Thank you, we have reviewed our data and reported the proportion of participants with ethnicity data derived from self-report in our results (lines 156-158). We have included this as a limitation of our study in our discussion (lines 309-311). 

4. The analyses are over-adjusted, as comorbidities and socioeconomic position (IMD) are potential mediators or drivers of ethnic inequalities in COVID-19 health outcomes. Ethnicity itself is not a cause of poor health outcomes, it is that ethnic minority groups are more likely to have comorbidities and lower socioeconomic position and this increases the risk of poor health outcomes. Though the authors do present unadjusted analyses in Table 2 (not stated in the methods), the authors should consider presenting only age and sex adjusted analyses as the primary analyses, then adjusting for comorbidities and socioeconomic position to determine whether controlling for factors attenuates differences between groups.

Reply: Thank you for your comment. We have updated the methods to state that we have performed unadjusted multiple regression analyses (line 123).

As recommended, we have performed age- and sex-adjusted analyses for follow-up symptoms and blood tests. We have updated our methods (line 123) and results accordingly. 

When reporting follow-up symptoms, we have first reported age- and sex-adjusted analyses and then analyses that adjust for comorbidities and socioeconomic position to determine if this attenuates differences between groups. We have updated Table 2 to include all analyses. We have also reviewed and updated our discussion accordingly. 

Similarly, we have included age- and sex-adjusted analyses when reporting follow-up blood tests. We have updated the relevant section in the results and discussion.

5. In the discussion, the authors state that they are unclear if the large confidence intervals are due to the small sample size or possible clinical significance, but confidence intervals are directly related to the size of the sample. The small sample size of this study limits the ability to determine reliable conclusions. Therefore, the authors should emphasize the need for larger and more representative studies to truly determine whether there are ethnic inequalities in long-COVID-19 symptoms.

Reply: Thank you for your comment. We have updated the line mentioned (lines 305-306) and updated our discussion to emphasise the need for larger and more representative studies. 

Reviewer #2

1. Page 5, final line, and page 14, line 293: please state the ethics review board by name.

Reply: Thank you, the study was approved by HRA and Health and Care Research Wales (HRA number 20/HRA/4928) and we have updated the manuscript as requested (lines 109-111). 

2. Page 6, lines 114-115: “continuous data were presented as medians with interquartile range (IQR) and categorical data as proportions”; for the ease of the reader, please remind them in the results section about this; e.g.

- Page 8, line 159: median age

- Page 8 line 162/163: median deprivation decile

Reply: Thank you, we have reviewed all results and specified where continuous data were presented as medians in lines 169, 172 and 173.

3. Page 7, lines 143-144, “All tests of significance were two-tailed and p-values ≤0.05 were considered statistically significant.” Would it be helpful to remind the reader that the Bonferroni correction is used? Did you use that for all testing? If not, please can you make it clearer in the methods where did and didn’t apply the Bonferroni correction?

- Same on page 9, line 180.

Reply: Thank you, Bonferroni corrections were applied for comparisons between continuous data using Kruskal-Wallis and categorical data using Fisher’s exact test, as mentioned in the methods. We have amended the wording in these lines (118-121) to make this clearer. 

We have also, as suggested, reminded the reader where Bonferroni correction is used in lines 149-150. 

In line with reviewer #1’s comments, we have amended the manuscript to describe first age- and sex- adjusted analyses, so page 9, line 180 (now line 197) no longer refers to analyses where Bonferroni correction was used. 

4. Page 7, line 148: space missing: “(66.7%)adults”

Reply: Thank you for noticing this, we have updated this (line 154). 

5. Page 8, line 158: “Post-hoc Bonferroni analyses”: this is usually called something like ‘post-hoc analyses adjusted using the Bonferroni correction’, please update the language.

Reply: We have updated the language as suggested (line 167). 

6. Page 9, line 180/181: suggest insertion of the word ‘overall’ to distinguish these results from the results looking at ethnic groups individually (in the supplementary table), i.e. “Post-hoc Bonferroni analyses showed no differences between ethnic groups in terms of unadjusted follow-up outcomes (Table 1).” To “Post-hoc Bonferroni analyses showed no differences between ethnic groups overall in terms of unadjusted follow-up outcomes (Table 1).”

Reply: As in comment (3), this line has now been removed. 

7. Page 9, line 193/194: “The only factor 194 which was correlated with number of other symptoms was presentation during alpha wave (alpha CR 4.13 (1.47 – 11.60), p = 0.01 vs. delta).” – In the same supplementary table, alpha wave was correlated with fatigue, and poor sleep quality was correlated with both the absence of hypertension and alpha wave. The reader may be interested in these, and I think they should be mentioned in the text as not all readers look at supplementary tables.

Reply: Thank you for highlighting this. We have updated the paragraph to include these factors (lines 217-221). 

8. Page 11, line 223: “Our data includes a small number of individuals (3.45%) who were clinically diagnosed with COVID-19. This rate compares favourably, for example to the RECOVERY trial which had a false-negative rate of approximately 10%21.” – please make it clearer in the text the distinction between this 3.45% and the other individuals in the study (e.g. positive test result). Please also make it clearer or update the statement about the RECOVERY trial – were these false-negative results? How was that determined?

Reply: Thank you for your comment. We have updated the text in the result section (lines 164-165) to clarify the distinction between this 3.45% and other individuals in the study. 

We have also updated the discussion in lines 265-267 to clarify the findings within the RECOVERY trial. 

9. Page 12, first paragraph: Great – really interesting and relevant discussion of the wider social context – thank you.

Tables: would you consider adding decimal places to some of the median values? Looking at Table 1,

- Age is presumably reported as median (IQR), with 2 decimal places.

- Deprivation decile: only one of the values has one decimal place – please make this more consistent – maybe with decimal places for all the median values reported, and for the IQR?

- Clinical frailty score: the medians are 3, 3, 2, 3, 3, yet the difference between groups is statistically significant. I think reporting decimal places would be useful for the reader. Additionally only two of the values of the IQRs have decimal places, both reported to two decimal places.

- Please go through the tables to check for this; in particular Supplementary Table 1, Covid-19 characteristics, admission number of symptoms; median 3 compared to a median also 3, and the difference is significant.

Reply: Thank you for your comment. We have reviewed all tables and updated all median and IQR values to two decimal places. We have also reviewed the data to confirm that differences between group are statistically significant. 

We have also reviewed post-hoc Bonferroni corrections and ensured these are accurate. 

10. I suggest you update the row header in tables from “Severity (moderate)” to “Acute COVID-19 Severity (moderate)” for the ease of the reader.

Reply: Thank you, we have updated this throughout the tables. 

11. Supplementary table 3, respiratory symptoms, ‘Other’: the result is non significant. Following the rest of your reporting I wouldn’t expect this result to be in bold – is it bold in error?

Reply: Thank you for this, we have updated the table.

---

## [Decision Letter · Decision Letter 1]

14 Oct 2024

Associations between ethnicity and persistent physical and mental health symptoms experienced as part of ongoing symptomatic COVID-19

PONE-D-24-27497R1

Dear Dr. Naidu,

We’re pleased to inform you that your manuscript has been judged scientifically suitable for publication and will be formally accepted for publication once it meets all outstanding technical requirements.

Kind regards,

Tai-Heng Chen, M.D., Ph.D.

Academic Editor

PLOS ONE

Reviewers' comments:

Reviewer's Responses to Questions

**Comments to the Author**

1. If the authors have adequately addressed your comments raised in a previous round of review and you feel that this manuscript is now acceptable for publication, you may indicate that here to bypass the “Comments to the Author” section, enter your conflict of interest statement in the “Confidential to Editor” section, and submit your "Accept" recommendation.

Reviewer #1: All comments have been addressed

2. Is the manuscript technically sound, and do the data support the conclusions?

Reviewer #1: Yes

3. Has the statistical analysis been performed appropriately and rigorously? 

Reviewer #1: Yes

4. Have the authors made all data underlying the findings in their manuscript fully available?

Reviewer #1: No

5. Is the manuscript presented in an intelligible fashion and written in standard English?

Reviewer #1: Yes

6. Review Comments to the Author

Reviewer #1: I thank the authors for taking the time to address the comments provided by the reviewers and editor. I have only a minor suggestion for an amendment in the abstract, to alter the phrasing from "Ethnicity can influence susceptibility to..." to "There is evidence of ethnic inequalities in...". It is not ethnicity itself that influences susceptibility to COVID-19 infection etc, but wider factors (as outlined by the authors in the introduction) that relate to ethnicity. After making this minor amendment, I believe this paper is suitable for publication.

7. PLOS authors have the option to publish the peer review history of their article (what does this mean?). If published, this will include your full peer review and any attached files.

Reviewer #1: **Yes: **Patricia Irizar

---

## [Editor Report · Acceptance letter]

23 Oct 2024

PONE-D-24-27497R1 

PLOS ONE

Dear Dr. Naidu, 

I'm pleased to inform you that your manuscript has been deemed suitable for publication in PLOS ONE. Congratulations! Your manuscript is now being handed over to our production team.

Kind regards, 

on behalf of

Dr. Tai-Heng Chen 

Academic Editor

PLOS ONE